

# Equatorial wave circulation associated with subseasonal convective variability over the subtropical western North Pacific in boreal summer

Peishan Chen[1,2,3], Katharina M. Holube[3], Frank Lunkeit[3], Nedjeljka Žagar[3], Yuan-Bing Zhao[3], and Riyu Lu[1,2]

[1]National Key Laboratory of Earth System Numerical Modeling and Application, Institute of Atmospheric Physics, Chinese Academy of Sciences, Beijing 100029, China
[2]College of Earth and Planetary Sciences, University of Chinese Academy of Sciences, Beijing 100049, China
[3]Meteorological Institute, Center for Earth System Research and Sustainability, Department of Earth System Sciences, University of Hamburg, Hamburg, 20144, Germany

**Correspondence:** Peishan Chen (chenpeishan20@mails.ucas.ac.cn)

**Abstract.** Atmospheric convection over the subtropical western North Pacific (SWNP) during boreal summer varies with a lifespan around 10 days, with significant effects on both local and remote circulation. Among the less understood effects is the coupling between SWNP convection variability and variability in equatorial wave circulation. This paper quantifies equatorial wave perturbations and their evolution throughout the SWNP convection lifespan, using wave space regression between out-going longwave radiation over the SWNP region and spectral expansion coefficients of global tropospheric circulation from ERA5 reanalyses. The regression distinguishes between convection-coupled Rossby and Kelvin waves, and mixed Rossby-gravity (MRG) and inertia-gravity (IG) waves. The former two correspond to the Gill solution of tropical wave response to asymmetric heating. The results show that MRG and IG waves exhibit amplitudes comparable to those of the Gill response component in the upper troposphere. In particular, MRG and IG waves dominate the cross-equatorial northerly flow over the Maritime Continent, with MRG waves being more important. These findings suggest caution in applying the Gill solution to interpret circulation responses to asymmetric heating sources in model simulations.

As SWNP convection intensifies, the MRG wave northerly winds across the equator strengthen, while IG waves represent enhanced upper-tropospheric outflow over the SWNP region. By contrast, the combined effect of Kelvin and Rossby waves reinforces circulation on the equatorward flank of the anticyclone over the SWNP region, with Rossby wave easterlies being about three times stronger than those associated with Kelvin waves. The Rossby wave signal resembles the $n = 1$ Rossby wave, with its Southern Hemisphere (SH) subtropical anticyclonic gyre forming over the southern Indian ocean during the decay phase of the SWNP convection. This subtropical Rossby wave gyre, combined with the IG meridional flow in the SH, acts as a bridge between the SWNP convection and extratropical circulation during austral winter.



# 1 Introduction

During boreal summer, the dominant feature of circulation over the subtropical western North Pacific (SWNP) is deep atmospheric convection, which acts as an off-equatorial heating source (e.g. Nitta, 1986). The effects of SWNP convection variability on the circulation and climate over the SWNP and Asia have been well documented (Nitta, 1987; Huang and Sun, 1992; Lau et al., 2000; Lu and Dong, 2001; Wang et al., 2001; Kosaka and Nakamura, 2006, and many others). Enhanced (suppressed) SWNP convection is associated with cyclonic (anticyclonic) circulation anomaly over the SWNP region in the lower

troposphere (Nitta, 1987; Huang and Sun, 1992; Lu and Dong, 2001; Wang et al., 2001; Kosaka and Nakamura, 2006, 2010; Chen and Lu, 2023). In the upper troposphere, the SWNP convection induces anomalous cross-equatorial flow over the Maritime Continent (Zhao et al., 2022). In addition, the SWNP convection is also associated with teleconnection patterns in both the Northern Hemisphere (NH) and Southern Hemisphere (SH), connecting the tropical and extratropical circulation anomalies (Nitta, 1987; Huang and Sun, 1992; Zhu et al., 2020; Lin, 2009; Zhao et al., 2019).

While most of above-cited studies focused on monthly or seasonal timescales, some studies also paid attention to the subseasonal variability of SWNP convection (Wang et al., 2016; Di Capua et al., 2020; Zhu et al., 2020; Sun et al., 2021; Sun and Tan, 2022; Quinting et al., 2024). For example, Sun et al. (2021) and Sun and Tan (2022) discussed the approximately 10-day long lifespan of the SWNP convection and the evolution of its associated Pacific-Japan pattern and Australia-South Pacific-South Atlantic pattern, whereas Zhao et al. (2023) provided climatology of cross-equatorial flow events over the Maritime Continent

in ERA5 reanalysis data (Hersbach et al., 2020) and connected it with the convection over the SWNP region. The enhancement of the SWNP convection is associated with the westward propagation of the anomalous convection from the equatorial central Pacific region and the cross-equatorial flow over the Maritime Continent appears in parallel with the convection moving away from the equator. Furthermore, the equatorial meridional wind anomalies have a baroclinic vertical structure and propagate westward.

The present study complements previous work on subseasonal variability of the SWNP convection during boreal summer by coupling its internal dynamics with equatorial waves. Previous studies discussing equatorial waves in relation to convective variability over the SWNP region include Chen and Sui (2010), Straub and Kiladis (2003), Feng et al. (2020a, b), and Ferrett et al. (2019, 2023). Based on qualitative analysis of a case study, Chen and Sui (2010) suggested that the 10-20 day oscillation in the convection over the SWNP during boreal summer is associated with circulation anomalies identified as the $n = 1$ equatorial

Rossby wave. Straub and Kiladis (2003) studied the boreal summer intraseasonal oscillation, which they filtered as the period of 30-96 days. They found that the evolution of convection oscillation between the tropical Indian Ocean and western Pacific, as defined by a convection within 7.5°-12.5° N and 100°-105° E, is correlated with the mixed Rossby-gravity (MRG) wave variability in the northern Indian Ocean and western Pacific, and Kelvin wave variation in the central Pacific. Feng et al. (2020a, b) discovered the characteristics of the rainfall progression in tropical depression (TD)-type waves over the SWNP and

found that this rainfall progression is controlled by horizontal moisture advection. Extreme precipitation over the Philippines in boreal winter is also related to equatorial wave activity, and it is found that the increased rainfall is coupled with the Rossby





and MRG wave activity (Ferrett et al., 2019). The forecast improvements of wave activity are argued to improve the heavy rainfall forecasts (Ferrett et al., 2023).

Although earlier studies have addressed various aspects of wave circulation associated with SWNP convection, a compre-
hensive analysis using reanalysis data is still lacking, both in terms of quantifying wave anomalies and elucidating the dynamics underlying the evolution of summer SWNP convection. In this study, we quantify the equatorial wave variance spectrum associated with SWNP convection during boreal summer, including MRG waves, Kelvin waves, equatorial Rossby waves and inertia-gravity (IG) waves. We examine the evolution of wave circulation throughout the lifespan of SWNP convection using a novel wave-convection regression methodology in wave space. In particular, we relate the evolution of cross-equatorial flow
during convective episodes to MRG and IG waves, while predominantly along-equatorial flow is associated with Kelvin and Rossby waves. The latter is known as "the Gill response" to localized off-equatorial heating.

Gill (1980) showed that the asymmetric forcing produces Rossby wave response to the west of the heating source with the cross-equatorial flow (the meridional wind) in the lower troposphere pointing towards the localized subtropical heating. The cross-equatorial flow consists of the Rossby modes with even meridional mode index seen as a pair of gyres with stronger flows
on their equatorward sides. The steady-state solution for asymmetric forcing is in response to enhanced convection north of the equator and suppressed convection of the equal magnitude and structure south of the equator, and it is commonly combined with the response to heating localized at the equator as a more representative solution for circulation response to the heating in one hemisphere (figure 3 in Gill, 1980). For example, convection in boreal summer is largely in the Northern NH subtropics whereas tropics-extratropical interactions are likely to take place in subtropical latitudes south of the equator in austral winter.
Gill's solution for the circulation response to a stronger heating north of the equator and cooling south of the equator consists of Rossby and Kelvin modes. Details of their spatial structure depend on the dissipation and other coefficients of the system (e.g. Wu et al., 2000; Schubert et al., 2009; Bellon and Reboredo, 2022; Reboredo and Bellon, 2022).

The analytical Gill's solution is based on the meridional geostrophy or the "long-wave approximation" (Pedlosky, 1965), also referred to as the "low-frequency approximation", which consists of neglecting the local time derivative in the meridional
momentum equation linearized about a state of rest (Eq. 12 in Gill, 1980). The approximation leaves the equatorial Kelvin wave unaffected and filters out both IG waves and MRG waves from the solutions (Boyd, 2018, Chapter 4). As demonstrated by our research, the long wave approximation limits usefulness of the Gill solution for asymmetric forcing and real flow configurations associated with seasonal cycle, synoptic-scale processes and interactions with extratropics. Importantly, this limitation has not been previously quantified, particularly at subseasonal scales, which are the subject of the present study.
We hypothesize that the cross-equatorial flow accompanying the lifespan of SWNP convection is associated with MRG and IG wave meridional winds that are not part of the Gill response to the asymmetrical heating. This cross-equatorial component of the response will be compared with the flow component made of Rossby and Kelvin waves. We aim at providing a comprehensive picture of equatorial waves during the evolution of SWNP convection in boreal summer by answering the following questions:

1. How does the equatorial wave circulation evolve in relation to the SWNP convection in boreal summer?





2. What roles do MRG and IG waves, in comparison to Rossby and Kelvin waves, play in different phases during the SWNP convection evolution?

To address these research questions, we first analyze observations of the outgoing longwave radiation (OLR), as a proxy of convection, to establish the SWNP convection index similar to previous studies. This is followed by the equatorial wave analysis using the fifth European reanalysis dataset ERA5 (Hersbach et al., 2020). Coupling between the equatorial waves and convection variability is established through regression in wave space. The method is described in section 2, along with data, with additional details and illustrations in Supplementary Information (SI). Results are presented in Section 3, and conclusions and outlook are given in section 4.

## 2  Data and Method

### 2.1  OLR data

We use daily mean OLR data from NOAA (Liebmann and Smith, 1996) for boreal summer (June to August; JJA) from 1979 to 2021 at a horizontal resolution of $2.5° \times 2.5°$. The SWNP convection index (SWNPI) is defined as the daily OLR anomaly averaged over the region between 10°N and 20°N, 110°E and 160°E and multiplied by -1. The anomaly is calculated as a departure from the 43-year mean for each day from June to August. The mean from all daily indexes is removed and the resulting deviations from the mean are divided by the standard deviation for each element. This definition follows earlier studies (e.g. Lu and Dong, 2001; Kobayashi et al., 2005; Xue and Fan, 2019; Chen and Lu, 2023). In this study, the sign is opposite to earlier studies in order to couple circulation patterns associated with positive values of SWNPI with circulation response to heating sources. Thus, a positive value of SWNPI denotes enhanced convection over the SWNP region, and vice versa.

The SWNPI has strong interannual and intraseasonal variability, which is well documented in previous studies (Nitta, 1987; Huang and Sun, 1992; Kosaka and Nakamura, 2006; Wang et al., 2016; Sun et al., 2021). The auto-correlation function of the daily SWNPI gives an e-folding timescale of about 5 days (Fig. S1), suggesting a lifespan of the SWNP convection being about 10 days, in agreement with Sun et al. (2021).

### 2.2  Wave decomposition of ERA5

Wave components of tropical circulation are obtained by carrying out modal decomposition of ERA5 reanalysis data (Hersbach et al., 2020) using the MODES software (Žagar et al., 2015). The MODES provides a multivariate representation of the global circulation in terms of 3D orthogonal vertical and horizontal structure functions (VSFs and HSFs, respectively). The VSFs are the numerical solutions of the vertical structure equation whereas the HSFs are eigensolutions of the spherical shallow-water equations, and are called the Hough harmonics. The Hough harmonics are defined as a product of the latitude-dependent Hough functions and harmonic waves in the longitudinal direction (e.g. Longuet-Higgins, 1968; Swarztrauber and Kasahara, 1985; Žagar and Tribbia, 2020). The horizontal and vertical structures are coupled by the eigenvalues of the vertical structure





equation, the so-called "equivalent depth". Additional information on the wave decomposition method is provided in SI and references therein.

Modal decomposition is wavenumber decomposition, applied at each time step with data without assumption on temporal
wave evolution. The decomposition produces time series of nondimensional complex expansion coefficients $\chi_\nu(t) = \chi_n^k(m;t)$. The 3-component index $\nu = (k,n,m)$ of $\chi_\nu$ consists of indices for the zonal wavenumber $k$, the meridional mode index $n$ and the vertical mode index $m$. The meridional indices differentiate between the linearly balanced (or nearly non-divergent) Rossby modes and linearly unbalanced IG modes. The IG modes consist of eastward- and westward-propagating solutions, denoted EIG and WIG respectively. We use 'modes' and 'waves' interchangeably but the latter refers to the case without the zonal mean
state ($k = 0$). The meridional index $n$ can take integer values starting from 0 and it defines the order of the associated Legendre functions which are used to represent the non-dimensional horizontal oscillations on the sphere (Swarztrauber and Kasahara, 1985). In MODES, the MRG mode is labelled as the lowest balanced mode or $n = 0$ Rossby mode and the Kelvin mode is the $n = 0$ EIG mode. Note that the spherical mode indexing is different from the solutions on the equatorial $\beta$-plane (Matsuno, 1966); in this case the Kelvin mode is denoted the $n = -1$ whereas the $n = 0$ EIG mode is called the eastward-propagating
MRG mode (see discussion in Boyd, 2018).

## 2.3  Decomposition setup and climatology

The ERA5 data are used as daily averages of circulation on 26 pressure levels from 1000 hPa to 125 hPa on the regular N128 Gaussian grid which includes 512 and 256 points in the longitudinal and latitudinal directions, respectively. Only the data in boreal summer season June-July-August (JJA) during 1979 to 2021 are used. The projection involves 200 zonal wavenumbers
and the zonal mean state ($k = 0$), 70 meridional modes for each of the three wave species (EIG, WIG and Rossby) and 19 vertical modes. The expansion is complete except for the vertical truncation which leaves the lower troposphere incompletely represented by the projection (e.g. Žagar et al., 2009). This means that the sum of Rossby, Kelvin, MRG, WIG and EIG modes in physical space corresponds to the inverse of the complete $\chi_\nu$ signal. Similarly, as the 3D normal modes are orthogonal, all moments can be evaluated in the wave and regime space of the Rossby and IG modes. The total global wave variance is a sum
of variances in individual wave components.

Figure 1 shows the JJA climatological horizontal winds at 150 hPa and 850 hPa levels split into components associated with the Rossby modes and the Kelvin waves − the Gill solution, and with the IG modes and MRG waves. Their sum corresponds to the total circulation. The Gill part (Figs. 1a,b) is primarily the zonal circulation, whereas the meridional circulation is predominantly contributed by the MRG and IG modes (Figs. 1c,d). Specifically, the Gill part largely contributes to the tropical
easterly jet in the upper troposphere (Figs. 1a,e) and the zonal flow of Somali jet in the lower troposphere (Figs. 1b,f). The MRG and IG waves − non-Gill part − plays an important role in providing the upper-troposphere northerly and lower-troposphere southerly flow across the equator over the tropical Indian Ocean and western Pacific (Figs. 1c-f), as hypothesized. In the SWNP region (blue box), the upper troposphere northerly outflow from the region across the Maritime Continent is represented by the MRG and IG modes whereas the north-easterly to easterly winds stand for the Gill part of the flow. Together, the two
components represent the north-easterly outflow over the deep summer convection in the region (Figs. 1a,c,e).





## 2.4 Coupling convection and wave circulation

In order to isolate wave modes related to the SWNP convection activity, we perform a linear regression between the time series of SWNPI and the complex expansion coefficient $\chi_n^k(m;t)$. Regression in wave space defined by the projection of 3D circulation onto different wave modes and selected circulation index was introduced by Žagar and Franzke (2015) who applied
the method to identify equatorial waves associated with the Madden-Julian Oscillation. The regression is computed as

$$\mathcal{R}_n^k(m,\tau) = \frac{1}{N-1} \frac{\sum_{t=1}^{N}\left[\left(\chi_n^k(m,t,\tau) - \overline{\chi_n^k(m)}\right)\left(SWNPI(t) - \overline{SWNPI}\right)\right]}{Var\left(SWNPI\right)}, \qquad (1)$$

where $\tau$ denotes the time lag, $\overline{(\ )}$ stands for time averaging and $Var\left(SWNPI\right)$ denotes the temporal variance of the SWNPI. The obtained complex coefficient $\mathcal{R}_n^k(m,\tau)$ describes the projection of the global circulation associated with the SWNP convection, as represented by the SWNPI, on Rossby, IG, Kelvin and MRG modes. The inverse projection of $\mathcal{R}_n^k(m,\tau)$ to physical
space provides the wind and geopotential anomalies associated with convective activity over the SWNP region.

To assess the relative temporal evolution of convection and waves, the regression is performed for $\tau$ varying from -6 days to +6 days, given that SWNP convection has the e-folding timescale of approximately 5 days. For $\tau < 0$, the OLR anomalies lag the circulation anomalies, and the opposite is true for $\tau > 0$. The discontinuity of the data in J-J-A during 1979 to 2021 is considered. Lead and lag days are defined relative to the calendar date of each index value. For the purpose of discussing the
dynamics, we assume enhanced convection (i.e., positive SWNPI) at day 0.

## 3 Equatorial wave circulation associated with SWNP convection

### 3.1 Circulation associated with SWNP convection at day 0

**Total circulation**

Figure 2 shows the total horizontal wind anomalies at 150 hPa and 850 hPa in the tropics associated with OLR variability over
the SWNP region at day 0. The upper-troposphere cross-equatorial northeasterly flow extending from the SWNP region to southeast Indian Ocean is related to enhanced SWNP convection. Its zonal component exceeds 5 m/s whereas the meridional components is locally over 3.5 m/s. This northeasterly flow, combined with the weaker southerly flow towards the extratropics, causes divergence over the SWNP region (Fig. 2a). In the lower troposphere, there is anomalous cyclonic circulation over the SWNP region with its southwesterly component crossing the equator over the Maritime Continent (Fig. 2b). The above-
described climatological circulation anomalies in boreal summer are similar to the results of previous studies, mostly applying for the monthly or seasonal timescale (Nitta, 1987; Lu and Dong, 2001; Wang et al., 2001; Li et al., 2019; Zhao et al., 2022; Chen and Lu, 2023).

Note that Fig. 2 is based on ERA5 data without involving modal decomposition; in other words, we computed correlations between the convection index and circulation in grid points at every level. An equivalent figure produced applying formula (1)
and inverse of the resulting complex field is provided in SI as Fig. S2. Deviations are very small, demonstrating that the MODES



decomposition is a complete representation of regional circulation. Now we proceed with the circulation decomposition to discuss how roles of different waves in shaping the observed anomalies in Fig. 2.

**Rossby and Kelvin waves**

Figure 3 presents the horizontal winds of Rossby waves and Kelvin waves associated with SWNP convection variability at two levels. Enhanced convection over the SWNP region is associated with significant easterly wind anomaly in Rossby wave flow over the Maritime Continent in the upper troposphere with the amplitude exceeding 3 m/s (Fig. 3a). The easterly wind anomaly over the Maritime Continent is additionally strengthened by the Kelvin waves (Fig. 3c), which contribute to the easterly outflow off the SWNP region extending from the tropical Maritime Continent to the Indian Ocean. Its amplitude exceeds 1 m/s. The Kelvin wave signal contributes to the upper-level divergence over the SWNP region, especially on its westerly side. The upper-tropospheric easterly wind anomaly at the equator is accompanied by the anomalous Rossby-wave anticyclonic circulation in the NH subtropics (Fig. 3a), especially over the SWNP region, as expected by the Gill solution. The centre of the anticyclone in the NH is over the SWNP box. The subtropical part of the Kelvin waves acts to weaken the Rossby-wave westerlies in the north and strengthen the Rossby-wave easterlies in the southern part of the SWNP region (Figs. 3a,c,e).

In the lower troposphere, the centre of anomalous cyclonic circulation is centred in the northern part of the SWNP region (Fig. 3b). The Rossby-wave westerlies on its south-western edge are combined with weaker Kelvin wave westerlies (Fig. 3d). The Kelvin wave signal can be seen up to and inside the SWNP region, this way enhancing the cyclonic circulation anomalies seen in the Rossby wave circulation. In summary, the Kelvin and Rossby wave flows act together to enhance the southern flank of cyclonic circulation over the SWNP region (Fig. 3f), resembling the well-known structure of the Gill solution for the circulation anomaly in response to asymmetric heating located north of the equator (Figs. 3e,f).

To quantitatively compare the contributions of Rossby and Kelvin wave component, we calculated the area-averaged zonal wind anomalies over 90°-150° E, 15° S-15 ° N at 150 hPa and 850 hPa. The Rossby and Kelvin wave contributions are 79% and 32% of the total average zonal flow at 150 hPa and 80 % and 11 % at 850 hPa, respectively. The sum of them are 111 % at 150 hPa and 91 % at 850 hPa. These indicate that other modes slightly suppress the upper-easterly and enhance the lower-westerly.

The two-level circulation picture is complemented with the full vertical structure of the zonal wind anomalies averaged over 15° S-15° N (Fig. 4). The longitude-pressure cross section for longitudes between 90° E and 150° E around the Maritime Continent shows the largest equatorial easterly flow at 150 hPa and the largest equatorial westerly flow at about 700 hPa, which is similar to the vertical structure of Kelvin waves, albeit the Kelvin wave signal is weaker than the Rossby waves (Figs. 4a-c), suggesting the reinforcing role of Kelvin waves in Rossby wave zonal wind. On the other hand, the latitude-pressure section shows that Rossby waves dominantly contribute to the asymmetric circulation anomalies between the NH and SH (Figs. 4d,e). Particularly, the lower-level Rossby waves in the SH are almost absent but the upper-level Rossby wave circulation is comparable to that in the NH, suggesting that the SWNP convection connects the circulation in the SH extratropics through the related upper-Rossby wave activity. By using numerical simulations, Goyal et al. (2021) also suggested that convection



over the Indo-Pacific warm pool as an asymmetric heating source causes a upper-level Rossby wave source in the subtropics to
215 excite a wave-train pattern in the SH extratropics.

**IG and MRG waves**

The horizontal winds for the IG and MRG waves are shown in Fig. 5 for the two mode types separately and as their sum.
It can be seen that both IG and MRG waves contribute to the cross-equatorial northerly flow from the SWNP region in the
upper troposphere (Figs. 5a,c,e). The strongest IG wave northerly winds exceed 2 m/s and are found at the southern flank of
220 the SWNP region. Together with the southerly flow towards the extratropics in the north of the SWNP region, the IG wave
circulation represents most of divergence over the SWNP region in the total signal in Fig. 2a, as could be expected (Neduhal
et al., 2023).

The overall strength of the northerly flow over the Maritime Continent is explained to a greater extent by the MRG waves
than by the IG waves. The MRG wave northerly winds extend further across the equator and reach the northern Australia with
225 amplitudes larger than 2 m/s (Fig. 5c). The zonal wind anomalies of the MRG waves are smaller than 1 m/s, but nevertheless
important for shaping the anomalous cyclonic and anticyclonic circulation to the west and east of the maximal MRG northerly
flow near 125° E, respectively, as a near-equatorial response to the enhanced SWNP convection. In the lower troposphere, both
MRG waves and IG waves contribute to the weak inflow over the SWNP region, with the former dominant and its southerly
flow crossing the equator over the Maritime Continent (Figs. 5b,d).

To quantitatively compare the contributions of IG and MRG wave components, we also calculated the area-averaged merid-
ional wind anomalies over 90°-150° E, 15°S-15 ° N at 150 hPa and 925 hPa. Considering that the maximum cross-equatorial
flow in the lower troposphere appears at 925 hPa (e.g. Zhao et al., 2023), the area-averaged anomalies at this level are calcu-
lated. The IG and MRG wave contributions at 150 hPa are 37% and 45% of the total meridional flow, respectively. At 925 hPa,
the percentages are 37% and 52% for the IG waves and MRG waves, respectively. This also indicates that the near-surface
meridional flow associated with SWNP convection is less balanced (i.e. projecting to a smaller extent to Rossby modes) than
in the upper troposphere (11% versus 18%).

Further details of the vertical structure of the MRG and IG wave meridional winds are presented in Fig. 6. It shows the
maximum northerly wind near 150 hPa, while nearly negligible southerly wind anomalies are found at 925 hPa, and the
meridional wind between 300 hPa and 700 hPa is almost absent, which is similar to that documented in Zhao et al. (2023).
The weak signal near the surface can be a result of the area averaged over 15° S-15° N or 90 °-150° E, as the cross-equatorial
southerly flow is featured by some branches across the Maritime Continent due to the topography (Fig. S3; Li and Li, 2014;
Zhao et al., 2023). The new aspect is the vertical structure decomposition showing that the two mode types have comparable
contributions in the upper troposphere, with the MRG wave component being stronger (Figs. 6a-c). Moreover, the latitude-
pressure cross section of the meridional wind anomalies exhibits asymmetry across the equator (Fig. 6d), which is attributed
to the dipole IG wave northerly flow over the north and south of the Maritime Continent (Fig. 6e and Fig. 5a).





The new insights provide an important departure from the Gill (1980) solution to the asymmetric forcing or to the sum of the symmetric and asymmetric forcing. We show that solutions based on the long-wave approximation ignore the cross-equatorial flow associated with outflow/inflow over regions with enhanced convection, which projects onto the IG and MRG waves.

### 3.2 Evolution of SWNP convection and associated equatorial wave circulation

Next, we discuss the results of lagged regression from day -6 to day 6 to examine the temporal evolution of the wave circulations in connection with the SWNP convection at day 0. For brevity, we focus on the flow at 150 hPa, where the circulation response is strongest.

**OLR and total circulation**

Figure 7 presents the regressed OLR and total horizontal wind anomalies at 150 hPa. Enhanced convection (negative OLR

anomalies) in the SWNP region at day -6 strengthens until day 0, followed by gradual weakening until day 6 (Figs. 7h-n). During this 12-day period, the centre of OLR anomaly moves westward from about 150° E at day -6 to 120° E at day 6 and slightly northward to the northern flank of the region. The relative movement is indicated by the red dot at the location of maximal convection.

As the SWNP convection grows stronger from day -6 to day 0, the northeasterly outflow from the region stretches across

the equator and expands westward while strengthening (Figs. 7a-d). Convection weakening after day 0 is accompanied by a westward shift of the northeasterly flow south of the equator and development of the anticyclonic circulation over the southern Indian ocean (Figs. 7e-g). The westward shift of cross-equatorial flow associated with the SWNP convection anomalies is consistent with Zhao et al. (2023).

In addition, the described evolution in the SH subtropics can be compared with Sun and Tan (2022), who analyzed the

265 evolution of the Australia-South Pacific-South Atlantic teleconnection pattern coupled with the subseasonal SWNP convection. This teleconnection arises from the upper-troposphere anticyclonic circulation over the Australia, which acts as a connection between the SWNP convection and SH extratropical circulation. An important result in Fig. 7 is the formation of the upper-troposphere anticyclonic circulation in the SH subtropics. Within this large-scale feature, southerly wind anomalies develop over the Australia and easterlies along the equator are enhanced, even after day 0. The circulation is very different from that at

270 the start of the convective cycle, and the northeasterly outflow from the SWNP region appears decoupled from the anticyclone in the SH. Additional understanding will be provided according to the roles of different equatorial waves.

**Rossby and Kelvin waves**

The decomposition reveals the relative roles of Rossby and Kelvin waves (Fig. 8) compared to IG and MRG waves (Fig. 9) in producing the circulation in Fig. 7. Figure 8 shows a growing Rossby wave signal as an upper-tropospheric anomalous

anticyclone over the convective region simultaneously with the intensification of SWNP convection (Figs. 8a-d). Moreover, the anticyclone in the SH also appears nearly simultaneously and keeps developing after day 0 (Figs. 8e-g). In between the two



anticyclonic gyres, there are easterly winds making the whole structure resemblant of the equatorial $n = 1$ Rossby wave. This is however not an analytical $n = 1$ Rossby wave - an eigensolution of the linear shallow-water equations on the equatorial $\beta$ plane (Matsuno, 1966) - but a structure modified by a complex interplay of convection, background flow and vertical stratification,
and filtered by the time-independent decomposition performed by MODES.

The $n = 1$ Rossby-wave-resembling anomaly slowly moves westward from the Maritime Continent towards Africa (Figs. 8a-g). The centre of easterly wind anomaly at the equator moves from about 140° E at day -6 to 110° E at day 6, which gives a westward phase speed of about 3.2 m/s. The OLR anomaly in Fig. 7 moves westward with a similar speed, but shifted about 10°to the east, in agreement with the Rossby-wave circulation response to tropical heating.

The Kelvin wave variability associated with the lifespan of SWNP convection is much less significant compared to the Rossby waves. At day 6, the Kelvin wave signal is nearly the same as that at day -6, except for the absence of westerlies east of 150° E at day 6 (Figs. 8h-n). Overall, this picture suggests that the Kelvin wave response to enhanced convective activity over the SWNP region does not significantly alter the pre-existing Kelvin wave easterlies over the eastern hemisphere, typical for the boreal summer (e.g. Žagar et al., 2022). There is a relatively local enhancement of equatorial easterlies along with the
SWNP convection from day -6 to day 0, accompanied with a slight westward movement. This slight westward shift of the Kelvin wave anomaly is due to the background easterlies (Fig. 1).

**IG and MRG waves**

The IG wave circulation is a part of upper-tropospheric outflow over the SWNP region, which is strengthening from day -6 until day 0. It consists of the northerly flow across the equatorial Maritime Continent and southerly flow in the subtropics
north of 15° N (Figs. 9a-d). As the SWNP convection weakens following day 0, the southerly winds from the east of Australia expand towards the equator and produce the upper-level convergence over the Maritime Continent (Figs. 9e-g). Another part of the IG signal is found in the southern Indian Ocean and moves westward quickly, which tends to be decoupled with the outflow over the SWNP region (Figs. 9e-g). It is associated with the northerly component of the total cross-equatorial circulation in Fig. 7, which in the Rossby wave part in Fig. 8 is more zonal. This is the cross-equatorial flow missing in the Gill part of the
circulation response to asymmetric heating. In the subtropics, the linear IG part of anticyclonic circulation is associated with the ageostrophic component of quasi-geostrophic dynamics (e.g. Žagar et al., 2023).

The upper-tropospheric MRG wave northerly winds across the equator also intensify along with the SWNP convection and are strongest at day 0 (Figs. 9h-k). For positive time lags (Figs. 9e-g), the MRG waves make a large part of the cross-equatorial northerly winds across the Maritime Continent while the part of IG waves is almost absent (Figs. 9l-n). Truthful to its definition
in linear theory, the MRG wave signal maximizes at the equator while propagating westward from about 149°E at day -6 to 102°E at day 6 (Figs. 9h-n). This speed (about 5 m/s) is significantly faster than that of $n = 1$ Rossby wave and might be associated with different shift due to the background easterlies and westward-moving convection.

The evolution of IG and MRG wave circulations at 925 hPa is provided in SI (Fig. S4) and it is similar to that at 150 hPa, i.e., the IG wave inflow over the SWNP region varies with the intensity of SWNP convection, while the MRG waves are a
310 dominant part of the westward-propagating, cross-equatorial southerly flow over the Maritime Continent. The convergence of





IG waves over the SWNP region at 925 hPa intensifies more rapidly than the convection there at day -2, which may indicate a positive feedback to the enhanced SWNP convection at later day 0 (Fig. S5). Then, both the divergence and convergence of IG waves at the higher and lower level reach the peak at day 0 as the enhanced convection.

Above results suggest different dynamics of IG waves and MRG waves. The IG waves couple with the enhanced SWNP convection through the upper-level outflow and lower-level inflow there. The upper-level outflow contributes to the cross-equatorial northerly flow over the Maritime Continent before day 0, but after that, the subtropical northerly flow in the SH tends to be decoupled with the outflow from the SWNP region, contributing to less cross-equatorial flow over the Maritime Continent. Moreover, this IG flow in the SH complements the Rossby wave-induced anticyclonic gyre over the southern Indian Ocean. On the other hand, the MRG waves may be excited by the asymmetric heating source, which moves westward along with the convection during its 10-day typical lifespan. As a result, the westward moving cross-equatorial flow over the Maritime Continent associated with the SWNP convection is dominantly caused by the MRG waves.

## 4 Conclusions and Outlook

Utilizing wave decomposition of ERA5 reanalyses and wave regression with OLR-based convection index, we investigated the roles of different equatorial waves in the evolution of atmospheric circulation anomalies associated with the convection over the subtropical western North Pacific (SWNP) region in boreal summer (JJA). In detail and regarding our initial questions, we found:

1. During its about 10-day lifespan, the convective activity over the SWNP enhances from day -6 to day 0, then gradually weakens until day 6, featured by a westward shift. The intensification of SWNP convection is associated with a Rossby wave signal resembling the n=1 mode with two developing anticyclonic gyres over the SWNP region and Australia and enhancing equatorial easterly flow between them in the upper troposphere. During the decay phase of SWNP convection, the anticyclone gyre in the Southern Hemisphere (SH) keeps extending westward. While the Kelvin wave variability is weaker compared to the Rossby waves, it nevertheless contributes to the equatorial easterly flow as a relatively localized response during the SWNP convection evolution. The inertia-gravity (IG) waves couple with enhanced convection through intensified outflow to the SH in the upper troposphere until day 0, and this outflow causes the cross-equatorial northerly flow over the Maritime Continent. After day 0, the IG northerly flow in the SH shifts westward to form the anticyclonic circulation over the southern Indian Ocean with the Rossby wave gyre together. Finally, the mixed Rossby–gravity (MRG) waves drive cross-equatorial meridional flow over the Maritime Continent during the period, propagating westward with a speed exceeding that of the associated SWNP convection.

2. The superposition of Kelvin and Rossby waves enhances equatorward side of the anticyclonic/cyclonic circulation over the SWNP region during stronger convection, with Rossby wave zonal winds being about three times stronger than those associated with Kelvin waves. Qualitatively, their zonal circulation resembles the Gill solution. On the other hand, the MRG and IG waves dominate the cross-equatorial northerly flow in the upper troposphere, and their amplitudes are

comparable to those of Rossby and Kelvin wave zonal circulations. Particularly, MRG waves contribute to a greater extent of the cross-equatorial northerly flow over the Maritime Continent and its westward propagation compared to IG waves.

Our results serve as a cautionary note against relying on the Gill solution to interpret circulation associated with asymmetric heating in observations, reanalyses, or weather and climate models. Future analyses should account for MRG and IG cross-meridional flows, which is not present in the original Gill framework. Although our study focus on the SWNP region, we anticipate similar dynamics associated with the subtropical convection in other regions.

One aspect of our results − extratropical circulation patterns associated with the SWNP convection − remains for further work. The presented results indicate the role of IG flow associated with the SWNP convection in shaping the anticyclonic circulation in the SH subtropics following peak convection. This complements earlier studies which noticed coupling between the SWNP JJA convection, the Rossby wave train in the SH and its influence on the SH rainfall (Lin, 2009; Zhao et al., 2019; Sun and Tan, 2022). Wave-space regression offers a novel approach to further investigate this coupling and the role of SWNP convection in austral winter variability.

*Data availability.* The ERA5 reanalysis data are openly available from https://cds.climate.copernicus.eu/datasets. The interpolated OLR data are provided by the NOAA/OAR/ESRLPSL, Boulder, Colorado, USA (https://psl.noaa.gov/data/gridded/data.olrcdr.interp.html). All data used in this work can be obtained from Peishan Chen through e-mail. The MODES software can be requested through https://modes.cen.uni-hamburg.de/software.

*Author contributions.* Peishan Chen: conceptualization, formal analysis, funding acquisition, investigation, validation, project administration, visualization, writing – original draft, writing – review and editing; Katharina Holube: formal analysis, writing – review and editing; Frank Lunkeit: conceptualization, writing – review and editing; Nedjeljka Žagar: conceptualization, formal analysis, funding acquisition, investigation, methodology, project administration, resources, software, supervision, visualization, writing – original draft, writing – review and editing; Yuan-bing Zhao: data curation; Riyu Lu: conceptualization, supervision, writing – review and editing.

*Competing interests.* The contact author has declared that none of the authors has any competing interests.

*Acknowledgements.* Peishan Chen gratefully acknowledges the financial support granted by the China Scholarship Council (CSC) Grant No. 202304910529. Nedjeljka Žagar acknowledges the Deutsche Forschungsgemeinschaft (DFG; German Research Foundation) Grants No. 461186383 and No. 274762653. Katharina M. Holube acknowledges the DFG Grant No. 461186383, and Yuan-Bing Zhao acknowledges the DFG Grant No. 274762653.



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





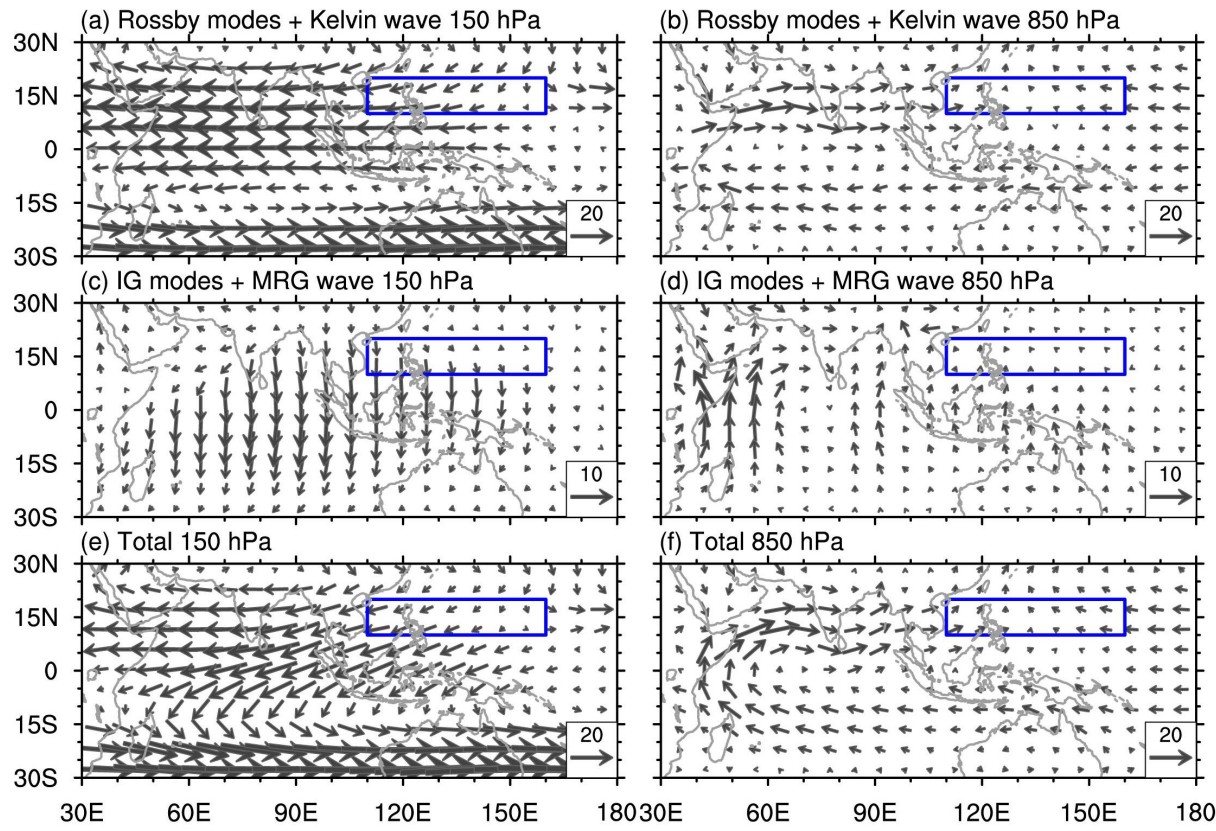

**Figure 1.** Climatological horizontal winds at (a,c,e) 150 hPa and (b,d,f) 850 hPa in boreal summer for (a-b) the sum of Rossby modes and Kelvin waves, (c-d) the sum of IG modes and MRG waves, and (e-f) total circulation. The blue box denotes the SWNP region.



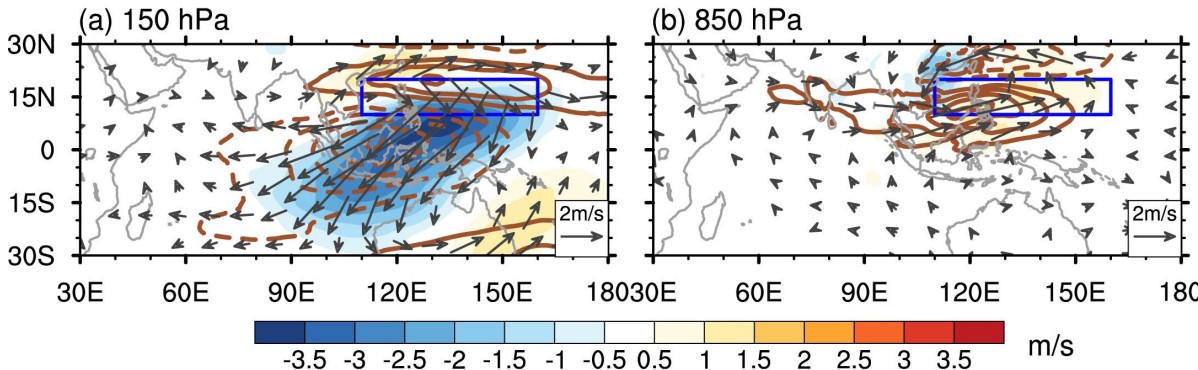

**Figure 2.** Total horizontal wind anomalies associated with the OLR variability over the SWNP region at (a) 150 hPa and (b) 850 hPa. Shading denotes the meridional wind whereas full and dashed contours represent the westerly and easterly zonal winds, respectively. The contouring interval for the zonal wind is ±1 m/s, and the zero contour is omitted. The contouring interval for the meridional wind is ±0.5 m/s starting at ±0.5 m/, as shown by the colorbar. Vectors show wind anomalies significant at the 95% confidence level based on the Student's *t* test. The blue box denotes the SWNP region.



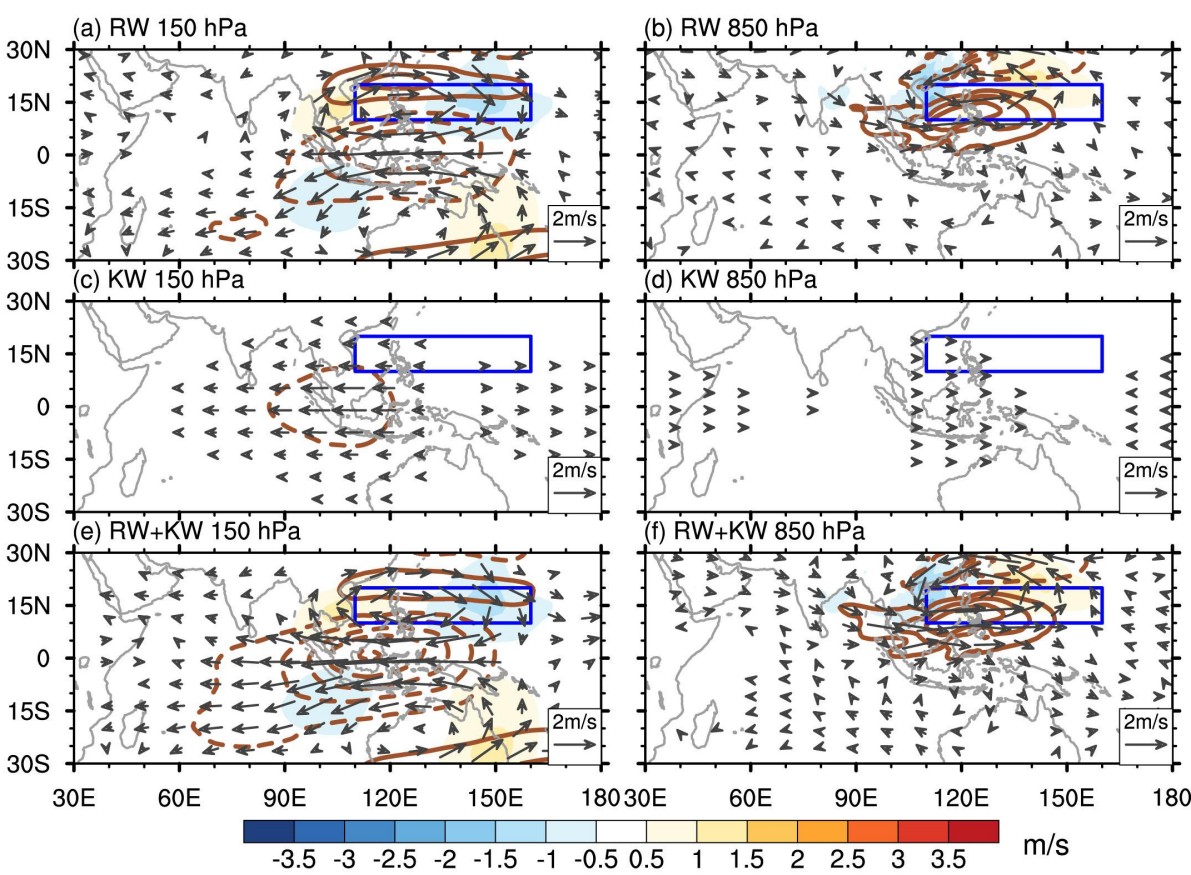

**Figure 3.** As in Fig. 2 but for (a-b) Rossby waves, (c-d) Kelvin waves and (e-f) the sum of Rossby waves and Kelvin waves associated with the OLR variability over the SWNP region at (a,c,e) 150 hPa and (b,d,f) 850 hPa. Vectors indicate the horizontal wind anomalies with magnitudes greater than 0.2 m/s and 0.1 m/s at 150 hPa and 850 hPa, respectively.

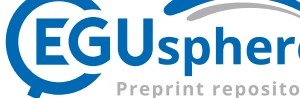

**Figure 4.** (a-c) Longitude-pressure cross section of the zonal wind perturbations between 40° E and 180° E, averaged over 15° S-15° N. (d-f) Latitude-pressure cross section of the zonal wind perturbations between 40° N and 40° S, averaged over 90° E-150° E. (a,d) the total signal, (b,e) Rossby waves and (c,f) Kelvin waves.




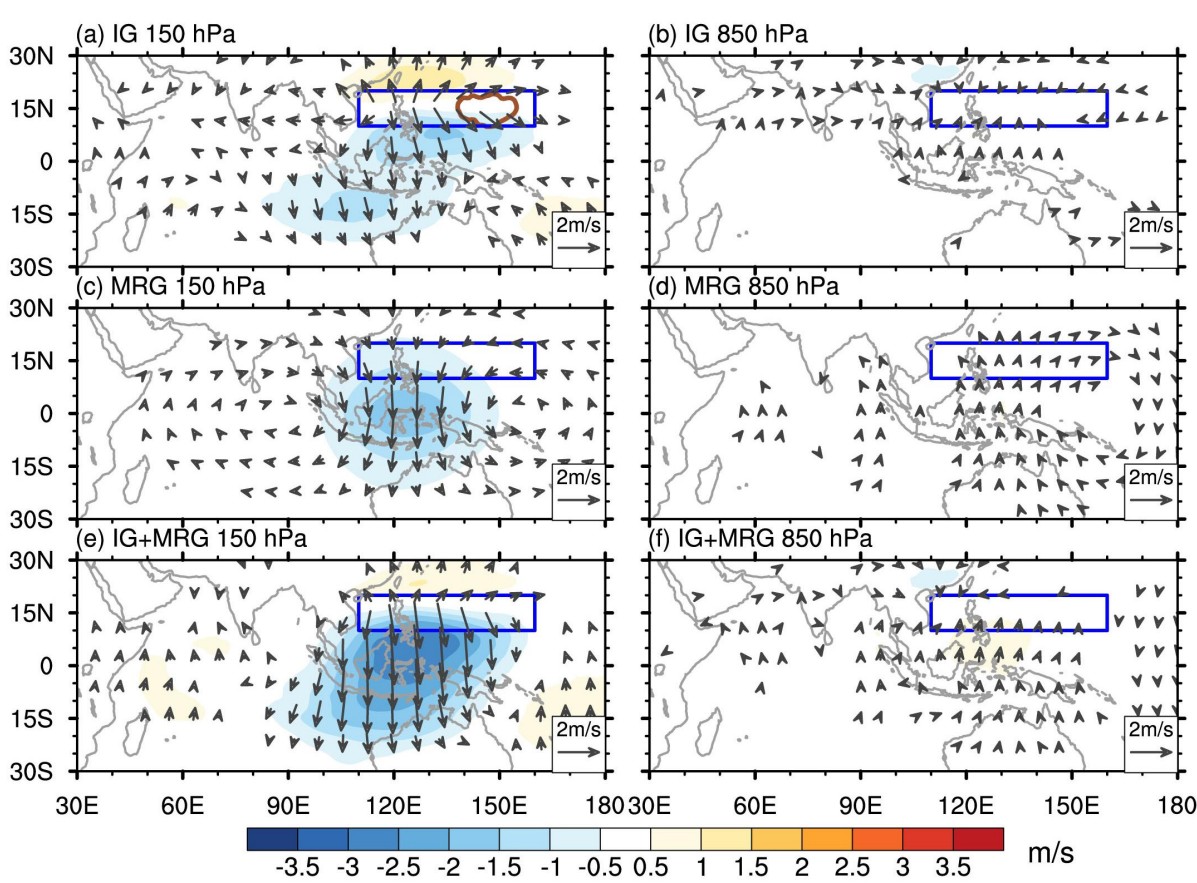

**Figure 5.** As in Fig. 3, but for (a-b) IG waves, (c-d) MRG waves and (e-f) the sum of IG waves and MRG waves.



**Figure 6.** As in Fig. 4, but for (b,e) IG waves, (c,f) MRG waves.



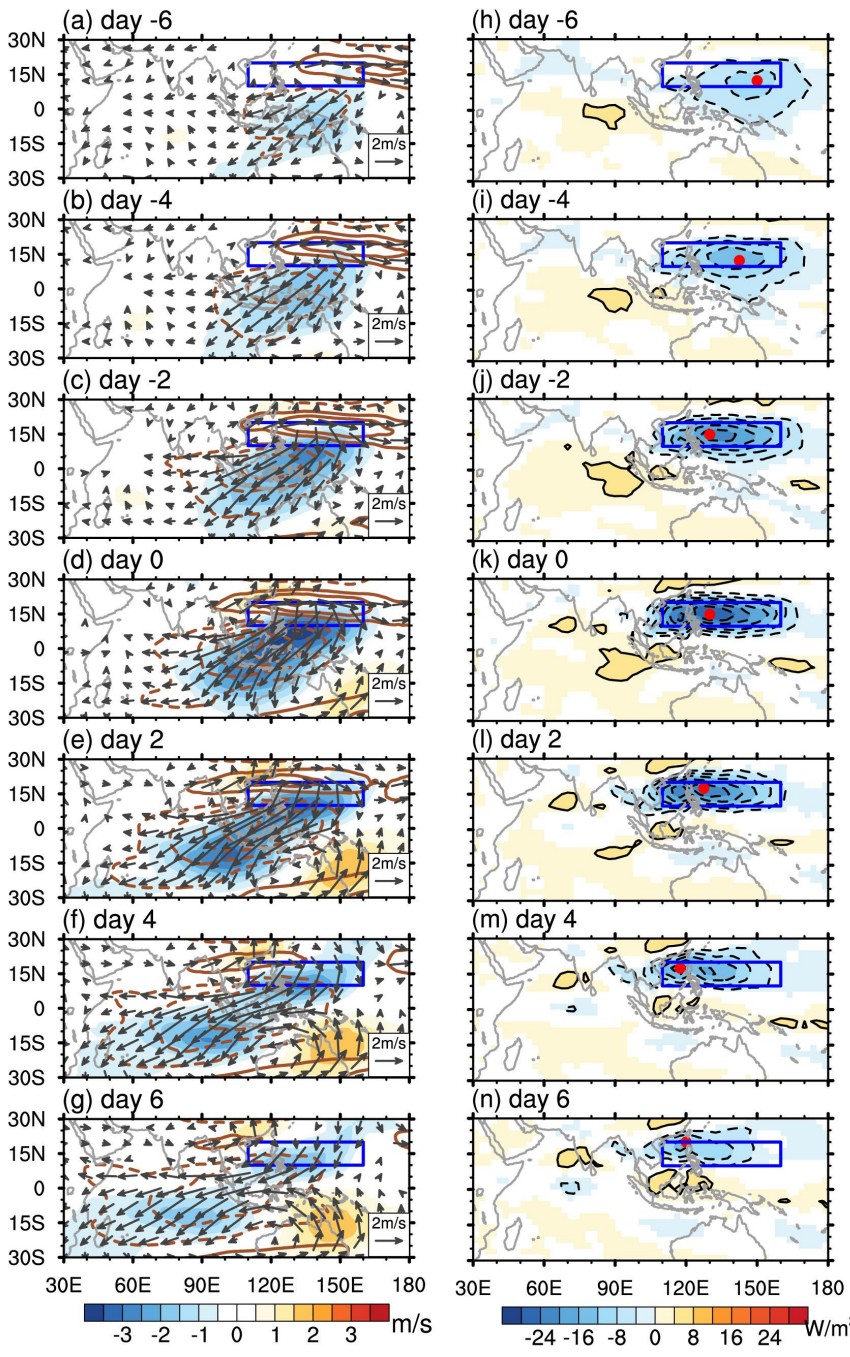

**Figure 7.** Results of the lagged regression between the circulation and OLR anomalies over the SWNP region. (a-g) total horizontal wind anomalies at 150 hPa (shading: v wind, contours: u wind), and (h-n) OLR anomalies. Significant OLR anomalies are shaded. The red dots denote the maximum OLR anomalies in the SWNP region. Day -6 to day -2 indicate that the OLR anomaly lags circulation from 6 to 2 days. Other details as in Fig. 3.



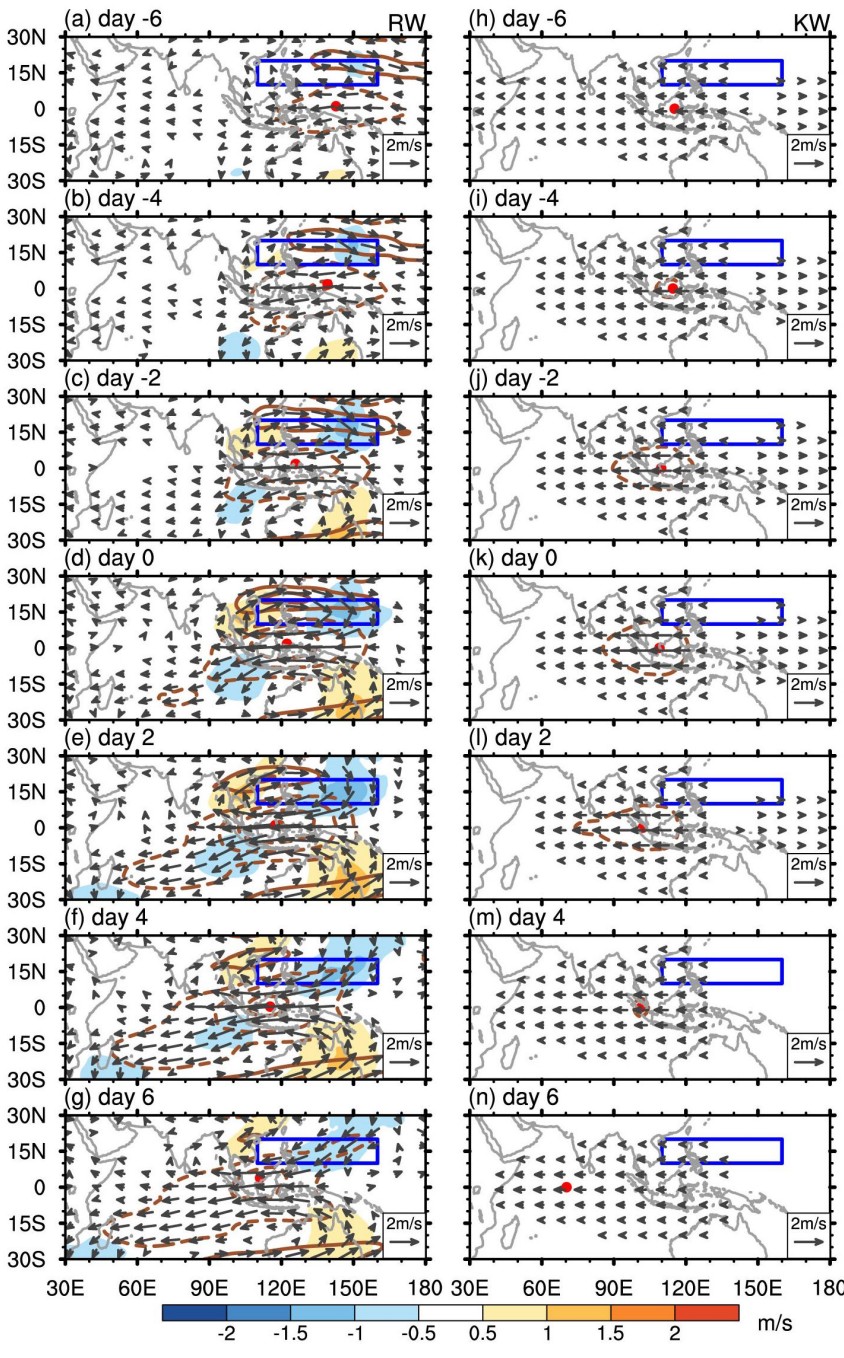

**Figure 8.** As in Fig. 7a-g, but (a-g) the Rossby wave part and (h-n) the Kelvin wave part of the horizontal wind anomalies. The red dots denote the maximum zonal wind anomalies over the 10° S-10° N and 40° E-160° E. To show the weak meridional wind anomalies, the color bars in Fig. 8 and Fig. 9 are different from that in Fig. 7. Other details as in Fig. 4.







**Figure 9.** As in Fig. 8, but for (a-g) IG waves and (h-n) MRG waves. The red dots denote locations of maximal meridional wind anomalies.