# Peer review of "Equatorial wave circulation associated with subseasonal convective variability over the subtropical western North Pacific in boreal summer"

_EGUsphere, 2025_

## Referee Comment (RC1)

Equatorial wave circulation associated with subseasonal convective variability over the subtropical western North Pacific in boreal summer
journal article review
en

**Review of egusphere-2025-2485**

Title: Equatorial wave circulation associated with subseasonal convective variability over the subtropical western North Pacific in boreal summer

**General comment**

The manuscript investigates the relationship between equatorial waves and convection over the subtropical western North Pacific (SWNP). This study applies wave decomposition to extract the signals of four equatorial waves and then regresses these onto OLR anomalies over the SWNP region. The authors provide a detailed analysis of the interaction between wave-related wind fields and SWNP convection. However, some viewpoints would benefit from further elaboration and discussion. Therefore, I recommend a major revision. My detailed comments are provided below.

**Specific comments**

1. Lines 17–18: Since MRG waves exhibit strong meridional flow across the equator (Fig. 9; Line 12), they might also contribute to linking SWNP convection with the extratropical circulation. Please consider discussing this possibility.

2. Lines 33–34: Please clarify what the "Pacific–Japan pattern" and the "Australia–South Pacific–South Atlantic pattern" refer to.

3. Lines 50–52: Since the manuscript mainly discusses boreal summer, citing studies focused on boreal winter here feels somewhat abrupt.

4. Section 2.2: Does the wave decomposition method allow for interactions among modes, or potential signal leakage between different wave modes?

5. Figure 1 / Lines 143–144: If the $u$ and $v$ components are shown separately as shading, can the discussion in Lines 143–144 still hold? Since the reference vector has different magnitudes for Rossby modes + Kelvin waves and IG modes + MRG waves, it is difficult to interpret this result as currently presented.

6. Lines 163–164: I do not understand the statement "the opposite is true for $\tau > 0$." Additionally, could the discontinuity or continuity in the data impact on the analysis or results?

7. Line 328: Why does convective activity over the SWNP region shift westward? Some explanation would help.

8. Figures (except Fig. 1): The blue box denoting the SWNP region is not very clear, especially when placed over the blue shading. Please consider improving its visibility.

9. Given that the manuscript emphasizes OLR anomalies as an indicator of SWNP convection, have the authors analyzed wave-related moisture, OLR, and precipitation in relation to SWNP convection? Such an analysis could strengthen the results.

**Minor comments**

1. Line 124: 'modes' and 'waves' → 'modes' and 'waves'

2. Line 179: Did the authors mean "at every pressure level"?

3. Line 265: It is not clear what is meant by the "Australia–South Pacific–South Atlantic pattern." Please clarify.

4. Lines 270–271: It is unclear how the authors determined that the northeasterly outflow from the SWNP is decoupled from the anticyclone in the SH.

5. Lines 287–289: The discussion would benefit from further elaboration on why pre-existing Kelvin waves are present over the SWNP.

6. Figure 1: The units for horizontal winds are not indicated.

7. Caption of Fig. 2: The current text reads "The contouring interval for the meridional wind is ±0.5 m/s starting at ±0.5 m/...". This seems to contain a typo. Should it be "The contouring interval for the meridional wind is ±0.5 m/s starting at ±0.5 m/**s**..."?

8. Caption for Fig. S2c–d: How did the authors compare Figures 3a–b with Figures 4a–b? Figures 4a–b show longitude–pressure cross sections, while Figures 3a–b are longitude–latitude maps.

---

## Referee Comment (RC2)

Review of "Equatorial wave circulation associated with subseasonal convective variability over the subtropical western North Pacific in boreal summer"

This is an interesting article about subtropical western North Pacific (SWNP) atmospheric convection in terms of wave activity and its coupling with convection. In particular, the roles of Rossby, Kelvin, MRG and IG modes are computed through normal mode decomposition techniques and highlights the roles of IG and MRG waves that are not present in Gill's original theoretical framework. The paper is generally well written and the scientific problem is clearly stated and deserves publication after a minor review. Particularly I would like to see more discussion on the relevant zonal wave numbers for SWNP convection for each mode type.

Minor issues.
lines 137-138: I did not understand the meaning of the phrase "This means that the sum of Rossby, Kelvin, MRG, WIG and EIG modes in physical space corresponds to the inverse of the complete χv signal.",

line 160: If I understand correctly formula (1) was not used in figure 1, how "horizontal wind anomalies at 150 hPa and 850 hPa in the tropics associated with OLR variability" is defined? I understand that figure 1 can be constructed from formula 1 by inverting it in physical space, but how was the figure originally constructed?

Line 200: Wouldn't this result also depend on the dominant zonal wavenumber of the Rossby and Kelvin waves (as large k components will average out close to zero and small k's with wavelengths larger than the box will have averages different from zero)? Do you have that information?

Figures 8 and 9: I was wondering if it is possible to summarize the finding of these two figures in a single figure showing some time series as a function of the lags? For instance the integrated values of $|u(t)|^2$ and $|v(t)|^2$ for the zonal and meridional velocities of each type of mode.

---

## Author Comment (AC3)

**Reviewer #1**

Dear Referee,

Thank you very much for your positive evaluation of our manuscript and your constructive comments and suggestions. Below please find your comments in bold font followed by our responses.

In addition, we have enclosed a draft of the revised manuscript, which incorporates the reviewers' comments, as detailed in the point-to-point responses.

Yours sincerely,

Peishan Chen, Katharina M. Holube, Frank Lunkeit, Nedjeljka Žagar, Yuan-Bing Zhao, and Riyu Lu

**Specific comments**

1. Lines 17–18: Since MRG waves exhibit strong meridional flow across the equator (Fig. 9; Line 12), they might also contribute to linking SWNP convection with the extratropical circulation. Please consider discussing this possibility.

**Response:** We agree and this is indeed one of the points of our discussion that we shall emphasize stronger in the revised paper. We have included a discussion on the possible contribution of the MRG waves to the link between SWNP convection and the extratropical circulation in Subsection 3.1:

"The upper-tropospheric MRG wave northerly winds across the equator also intensify along with the SWNP convection, and are strongest at day 0 (Figs. 9h-k). The cross-equatorial flow of the MRG waves, combined with the IG waves, links with the northerly anomalies on the westward flank of the Rossby wave anticyclone over Australia. This acts as an upper-level outflow extending from the SWNP region to the south-eastern Indian Ocean (Figs. 8a-d and Figs. 7a-d). For positive time lags (Figs. 9e-g), the MRG waves account for a large part of the cross-equatorial northerly winds across the Maritime Continent, while the contribution of IG waves is almost absent (Figs. 9l-n). The linkage between the MRG wave cross-equatorial flow over the Maritime Continent and the Rossby wave anticyclone in the subtropical SH gradually weakens (Figs. 8e-g), while the IG flow in the southern Indian Ocean moves westward, coupling

with the westward-extending Rossby waves (Figs. 9e-g)."

**2. Lines 33–34: Please clarify what the "Pacific-Japan pattern" and the "Australia-South Pacific-South Atlantic pattern" refer to.**

**Response:** Thank you for your question. We have clarified these by revising the introduction:

"For example, Sun et al. (2021) discussed the approximately 10-day long lifecycle of the SWNP convection along with the evolution of the Pacific-Japan pattern, which is characterised by a meridional dipole pattern over the north of the Philippines and east of Japan (e.g. Nitta, 1987; Zhu et al., 2020). Sun and Tan (2022) suggested a SH teleconnection of the SWNP convection over Australia, South Pacific and South Atlantic, which defines the Australia–South Pacific–South Atlantic pattern."

**3. Lines 50–52: Since the manuscript mainly discusses boreal summer, citing studies focused on boreal winter here feels somewhat abrupt.**

**Response:** We agree with your comment. Our intention was to give an as complete as possible overview of studies dealing with the equatorial waves and subtropical circulation in the region to indicate that the questions addressed in our paper have not been addressed by previous studies or other methods. This is why we would like to retain the citations, but to not elaborate on details.

**4. Section 2.2: Does the wave decomposition method allow for interactions among modes, or potential signal leakage between different wave modes?**

**Response:** Thank you for your question. The wave decomposition method could be applied to analyze modes interactions, and such a study has been carried out recently for the Kelvin wave over the global tropics by, e.g., Holube et al. (2025). They calculated dynamical Kelvin wave energy tendencies and it is possible to compare the results between wave-wave and wave-mean flow interactions. While we don't address modes interactions in this study, it is a possible extension of this work, to be considered in the future. We have added to the conclusions:

"Furthermore, while our study investigates the individual behavior of the different wave types, the normal-mode framework also allows for the more detailed analysis of the wave-wave and wave-mean flow interactions (e.g., Teruya et al., 2024). In this respect, the methodology applied by Holube et al. (2025) to Kelvin waves offers a starting point for further investigations on wave dynamics related to SWNP convection."

5. Figure 1 / Lines 143–144: If the u and v components are shown separately as shading, can the discussion in Lines 143–144 still hold? Since the reference vector has different magnitudes for Rossby modes + Kelvin waves and IG modes + MRG waves, it is difficult to interpret this result as currently presented.

**Response:** Thank you for your detailed comments on this figure. We have replaced Figure 1, which now includes shading and contours for the meridional and zonal wind component, respectively (see Fig. A1 below). The modified Figure confirms that the Rossby modes and the Kelvin waves primarily form the zonal circulation, whereas the meridional circulation is predominantly contributed by the MRG and IG modes as we state in the manuscript.

Fig.A1. Climatological horizontal winds at (a,c,e) 150 hPa and (b,d,f) 850 hPa in boreal summer

for (a–b) the sum of Rossby modes and Kelvin waves, (c–d) the sum of IG modes and MRG waves, and (e–f) the total circulation. The blue box denotes the SWNP region. Shading denotes the meridional wind whereas full and dashed contours represent the westerly and easterly zonal winds, respectively. The contouring interval for the zonal wind is  $\pm 15$  m/s starting at  $\pm 10$  m/s, and the zero contour is omitted. The contouring interval for the meridional wind is  $\pm 2$  m/s starting at  $\pm 2$  m/s, as shown by the colorbar.

6. Lines 163–164: I do not understand the statement "the opposite is true for  $\tau > 0$ ." Additionally, could the discontinuity or continuity in the data impact on the analysis or results?

**Response:** For clarification, we have revised this statement, which now reads:

"For  $\tau < 0$ , the OLR anomalies lag the circulation anomalies, and  $\tau > 0$  means that the OLR anomalies lead the circulation anomalies."

In addition, we note that the lead/lag days are defined as specific calendar days relative to day 0 in each individual year, and the false continuity between the data in August and the data in June in the next year is avoided. The statement on discontinuity was indeed confusing for the reader and we have deleted it in the revised manuscript.

7. Line 328: Why does convective activity over the SWNP region shift westward? Some explanation would help.

**Response:** Thank you for your question. Referring to previous studies, we have added:

"During this 12-day period, the centre of OLR anomaly shifts westward from about 150°E at day -6 to 120°E at day 6 and moves slightly northward to the northern flank of the region. This behaviour is consistent with the quasi-biweekly oscillation described by Chatterjee and Goswami (2004); Kikuchi and Wang (2009); Jia and Yang (2013). The relative movement is indicated by the red dots, which denote the location of maximum convection in the SWNP region at this particular day. This westward shift can be attributed to the zonal gradient of moisture anomalies and the mean easterly

trade winds, which together precondition moisture anomalies west to the convection (Li et al., 2020)."

and:

"The OLR anomaly in Fig. 7 moves westward with a similar speed, but shifted about 10° to the east, in agreement with the Rossby-wave circulation response to subtropical heating in earlier studies by (Chatterjee and Goswami, 2004) and (Liu et al., 2015)."

8. Figures (except Fig. 1): The blue box denoting the SWNP region is not very clear, especially when placed over the blue shading. Please consider improving its visibility.

**Response:** To improve visibility, we have used another color for this box in all respective figures.

•

9. Given that the manuscript emphasizes OLR anomalies as an indicator of SWNP convection, have the authors analyzed wave-related moisture, OLR, and precipitation in relation to SWNP convection? Such an analysis could strengthen the results.

**Response:** The modal decomposition used in this study provides a multi-variate representation of the global circulation including horizontal wind and geopotential height. We use OLR as an index of SWNP convection to study the convection-related wave circulation by regression as explained in Section 2 (Data and Method). Wave-related moisture, OLR and precipitation are not part of the wave decomposition and therefore the suggested analysis cannot be carried out using our methodology. However, with respect to your suggestion, we have added references to strengthen the role of MRG and IG wave cross-equatorial flow on moisture:

"Specifically, the IG wave inflow over the SWNP region varies with the intensity of SWNP convection, while the MRG waves constitute the dominant component of the westward-propagating, cross-equatorial southerly flow over the Maritime Continent. This flow has been identified as important for water vapor transport and for modulating precipitation over the SWNP region during boreal summer (Kubota et al., 2011; Li et al., 2018)."

**Minor comments**

1. Line 124: 'modes' and 'waves' → 'modes' and 'waves'

**Response:** Followed.

2. Line 179: Did the authors mean "at every pressure level"?

**Response:** This is "at 150 and 850 hPa" and we have revised it.

3. Line 265: It is not clear what is meant by the "Australia-South Pacific-South Atlantic

pattern." Please clarify.

**Response:** We have included the definition of the Australia–South Pacific–South Atlantic pattern

in response to Specific Comment 2 (see above).

4. Lines 270–271: It is unclear how the authors determined that the northeasterly outflow

from the SWNP is decoupled from the anticyclone in the SH.

**Response:** To clarify, we have modified the respective sentence:

"As the anticyclone expands westward towards the western Indian Ocean, the northeasterly

outflow from the SWNP region becomes increasingly separated from the northerly flow on the

westward flank, indicating a gradual decoupling between tropical and subtropical circulation."

5. Lines 287–289: The discussion would benefit from further elaboration on why pre-existing

Kelvin waves are present over the SWNP.

Response: The relative role of dynamical and diabatic forcing for this pre-existing, stationary

Kelvin wave is a subject of ongoing research, and we have added a reference:

"The relative role of dynamical and diabatic forcing for this stationary Kelvin wave is a subject of

ongoing research (Holube et al., 2025)."

6. Figure 1: The units for horizontal winds are not indicated.

**Response:** The units have been added accordingly.

7. Caption of Fig. 2: The current text reads "The contouring interval for the meridional wind

is  $\pm 0.5$  m/s starting at  $\pm 0.5$  m/...". This seems to contain a typo. Should it be "The contouring

interval for the meridional wind is  $\pm 0.5$  m/s starting at  $\pm 0.5$  m/s. . . "?

Response: Followed.

8. Caption for Fig. S2c-d: How did the authors compare Figures 3a-b with Figures 4a-b? Figures 4a-b show longitude-pressure cross sections, while Figures 3a-b are longitude-latitude maps.

**Response:** This caption has been revised to "(c–d) differences between Figs. 2a–b and Figs. S2a–b".

**References**

- Chatterjee, P. and Goswami, B. N.: Structure, genesis and scale selection of the tropical quasi-biweekly mode, Q. J. Roy. Meteor. Soc., 130, 1171–1194, https://doi.org/10.1256/qj.03.133, 2004.
- Holube, K. M., Lunkeit, F., Vasylkevych, S., et al.: Energy budget of the equatorial Kelvin wave: Comparing dynamical and diabatic sources. ESS Open Archive. https://doi.org/10.22541/essoar.174776017.78570412/v1. 2025.
- Jia, X. and Yang, S.: Impact of the quasi-biweekly oscillation over the western North Pacific on East Asian subtropical monsoon during early summer, J. Geophy. Res.-Atmos., 118, 4421–4434, https://doi.org/10.1002/jgrd.50422, 2013.
- Kikuchi, K. and Wang, B.: Global Perspective of the Quasi-Biweekly Oscillation, J. Clim., 22, 1340–1359,https://doi.org/10.1175/2008JCLI2368.1, 2009.
- Kubota, H., Shirooka, R., Hamada, J.-I., et al.: Interannual Rainfall Variability over the Eastern Maritime Continent, J. Meteorol. Soc. Jpn, 89A, 111–122, https://doi.org/10.2151/jmsj.2011-A07, 2011.
- Li, K., He, Q., Yang, Y., et al.: Equatorial Moisture Dynamics of the Quasi-Biweekly Oscillation in the Tropical Northwestern Pacific during Boreal Summer, Geophys. Res. Lett., 47, e2020GL090929, https://doi.org/10.1029/2020GL090929, 2020.
- Li, S., Park, S., Lee, J. Y., et al.: Chemical evidence of inter-hemispheric air mass intrusion into the Northern Hemisphere mid-latitudes, Sci. Rep., 8, 4669, https://doi.org/10.1038/s41598-018-22266-0, 2018.
- Liu, F., Huang, G., and Yan, M.: Role of SST meridional structure in coupling the Kelvin and Rossby waves of the intraseasonal oscillation, Theor. Appl. Climatol., 121, 623–629, https://doi.org/10.1007/s00704-014-1266-0, 2015.
- Nitta, T.: Convective Activities in the Tropical Western Pacific and Their Impact on the Northern Hemisphere Summer Circulation, J. Meteorol. Soc. Jpn. Ser. II, 65, 373–390, https://doi.org/10.2151/jmsj1965.65.3\_373, 1987.
- Straub, K. H. and Kiladis, G. N.: Interactions between the Boreal Summer Intraseasonal Oscillation and Higher-Frequency Tropical Wave Activity, Mon. Weather Rev., 131, 945–960, https://doi.org/10.1175/1520-0493(2003)131<0945:IBTBSI>2.0.CO;2, 2003.
- Teruya, A. S. W., Raphaldini, B., Raupp, C. F. M., et al.: Data-driven modeling of equatorial atmospheric waves: The role of moisture and nonlinearity on global-scale instabilities and propagation speeds, Chaos, 34, https://doi.org/10.1063/5.0201716, 2024.
- Zhu, Y., Wen, Z., Guo, Y., Chen, R., Li, X., and Qiao, Y.: The characteristics and possible growth mechanisms of the quasi-biweekly Pacific-Japan teleconnection in Boreal Summer, Clim. Dynam., 55, 3363–3380, https://doi.org/10.1007/s00382-020-05448-3, 2020.

---

## Author Comment (AC4)

**Reviewer #2**

Dear Dr. Raphaldini,

Thank you very much for your positive evaluation of our manuscript and your constructive comments and suggestions. Below please find your comments in bold font followed by our responses.

In addition, we have enclosed a draft of the revised manuscript, which incorporates the reviewers' comments, as detailed in the point-to-point responses.

Yours sincerely,

Peishan Chen, Katharina M. Holube, Frank Lunkeit, Nedjeljka Žagar, Yuan-Bing Zhao, and Riyu Lu

**Minor issues.**

1. lines 137–138: I did not understand the meaning of the phrase "This means that the sum of Rossby, Kelvin, MRG, WIG and EIG modes in physical space corresponds to the inverse of the complete xv signal.".

**Response:** Thank you for your comment. We have revised this sentence in the manuscript to make it clearer:

"This means that the original physical field corresponds to the inverse transform of all modes (the sum of Rossby, Kelvin, MRG, WIG, and EIG modes) from wave space to physical space."

2. line 160: If I understand correctly formula (1) was not used in figure 1, how "horizontal wind anomalies at 150 hPa and 850 hPa in the tropics associated with OLR variability" is defined?

I understand that figure 1 can be constructed from formula 1 by inverting it in physical space, but how was the figure originally constructed?

Response: You are right, the horizontal winds in Fig. 1 are climatological winds of the total

circulation (panels e and f), the sum of Rossby modes and the Kelvin waves (a and b), and the sum of IG modes and MRG waves (c and d), without regression (Eq. 1). For the different modes, the climatological winds are obtained by applying the inverse transform from wave space to physical space for the mean of the mode coefficients in JJA during 1979–2021. Figure 1 aims to visually introduce the climatological features of these modes and illustrate the 3D normal mode decomposition method.

We have added to the description of Fig.1:

"The climatological winds are obtained by applying the inverse transform from wave space to physical space for the time-averaged spectral expansion coefficients in JJA during 1979–2021."

3. Line 200: Wouldn't this result also depend on the dominant zonal wavenumber of the Rossby and Kelvin waves (as large k components will average out close to zero and small k's with wavelengths larger than the box will have averages different from zero)? Do you have that information?

Response: We fully agree with you that the Rossby and Kelvin waves with small zonal wavenumbers are dominant. We have analyzed the global variance of Rossby and Kelvin waves with different zonal wavenumbers associated with the SWNPI (Fig. A1), and the modes with small wavenumbers have larger global variance. Fig. A2 shows the horizontal circulation of k=1-3 Rossby and Kelvin waves, which are generally referred to as planetary scale waves. The results show that the amplitude of zonal wind anomaly in k=1-3 Rossby waves is about 76% of the total Rossby waves (Figs. A2a,b and Figs. 3a,b), and that of the k=1-3 Kelvin waves nearly contribute 78% to the total Kelvin waves (Figs. A2c,d and Figs. 3c,d). Therefore, the sum of k=1-3 Rossby

and Kelvin waves (Figs. A2e,f) resembles the pattern shown in Figs. 3e and 3f.

We have included Fig. A1 and Fig. A2 in the SI (new Figs. S6 and S7), and we have added to Section 3.1:

"This result is mainly contributed by planetary-scale (k = 1-3) waves, which exhibit a similar pattern to the total Rossby and Kelvin wave signals (Figs. S6 and S7)."

Figure A1. Global variance of Rossby waves and Kelvin waves associated with the SWNPI as a function of zonal wavenumber.

Figure A2. Horizontal wind anomalies of k=1-3 (a-b) Rossby waves, (c-d) Kelvin waves and (e-f) the sum of Rossby waves and Kelvin waves at 150 hPa and 850 hPa associated with the OLR variability over the SWNP region. Shading denotes the meridional wind whereas full and dashed contours represent the westerly and easterly zonal winds, respectively. The contouring interval for the zonal wind is  $\pm 1$  m/s, and the zero contour is omitted. The contouring interval for the meridional wind is  $\pm 0.5$  m/s starting at  $\pm 0.5$  m/s, as shown by the colorbar. Vectors indicate the horizontal wind anomalies with magnitudes greater than 0.2 m/s and 0.1 m/s at 150 hPa and 850 hPa, respectively. The blue box denotes the SWNP region.

4. Figures 8 and 9: I was wondering if it is possible to summarize the finding of these two figures in a single figure showing some time series as a function of the lags? For instance the integrated values of  $|u(t)|^2$  and  $|v(t)|^2$  for the zonal and meridional velocities of each type of mode.

**Response:** According to your suggestion, we calculate the U2 of Rossby waves and Kelvin waves, V2 of IG waves and MRG waves averaged over the Maritime Continent, and kinetic energy of Rossby and IG waves averaged over the southwest Indian Ocean to make a summary of Figs. 8

and 9 (Fig. A3). We have added figure A3 to the revised manuscript (new Figure 10), and we have added to the text (Lines 331-336):

"These results are summarized in Fig. 10 comparing time-longitude diagrams of the Rossby and Kelvin wave, IG and MRG wave, and the SH Rossby and IG wave energies. It shows that the Rossby waves exhibit a westward shift and play a dominant role in the equatorial zonal wind anomalies compared with Kelvin waves (Fig. 10a). On the other hand, Fig. 10b highlights the leading contribution of MRG waves to the westward-propagating cross-equatorial flow over the tropical Maritime Continent. Furthermore, both IG waves and Rossby waves over the southern Indian Ocean after day 0 are shown in Fig. 10c."

Figure A3. Time-longitude cross sections of different modes at 150 hPa. (a)  $U^2$  of Rossby (red Lines) and Kelvin (black lines) waves averaged over  $15^{\circ}S-15^{\circ}N$ , (b)  $V^2$  of IG (black lines) and MRG (red lines) waves over  $0^{\circ}$ , (c) Kinetic energy of Rossby (red lines) and IG (black lines) waves averaged over  $20^{\circ}S-0^{\circ}$ , respectively. The contouring interval is  $1 \text{ m}^2/\text{s}^2$  for (a) and is  $0.5 \text{ m}^2/\text{s}^2$  for (b) and (c).